# Effect of Financial Frictions on Monetary Policy Conduct: A Comparative Analysis of DSGE Models with and without Financial Frictions

**Salha Ben Salem** [1,*] **, Sonia Sayari** [2,*] **and Moez Labidi** [3]

1 Financial Development and Innovation (DEFI) Research Unit, University of Monastir, Monastir 5000, Tunisia
2 Department of Business Administration, College of Administrative and Financial Sciences, Saudi Electronic University, Ryiadh 13316, Saudi Arabia
3 Arab Planning Institute, Kuwait City 13059, Kuwait; moezlabidi@api.org.kw
* Correspondence: salha.bensalem@fsegma.u-monastir.tn (S.B.S.); s.sayari@seu.edu.sa (S.S.)

**Abstract:** In this study, we explored the impact of bank leverage and financial frictions on the transmission of real and financial shocks. Two new Keynesian dynamic stochastic general equilibrium (DSGE) models, with and without financial frictions, were employed in the context of the Tunisian economy. In the analysis, we considered three types of shocks—productivity, monetary, and adverse bank capital shocks. The findings reveal that, in the model with financial frictions, the response of macroeconomic and financial variables to demand and supply shocks was more pronounced than in the baseline model, where frictions primarily exist at the borrower level. In this study, we underscored the significance of financial shocks, particularly negative bank capital shocks, in triggering substantial macroeconomic and financial fluctuations, especially when banks operate with higher leverage ratios. Additionally, the inclusion of financial frictions in the DSGE model enhanced its ability to capture the empirical features of real and financial shocks, providing valuable insights for effective monetary policymaking. The results provide foundational insights for Tunisian policymakers to assess the impact of financial frictions in the context of the Tunisian economy. This is significant for the Central Bank of Tunisia, which has not yet adopted a specific DSGE model. Therefore, through our analysis, we determined the amplificatory role of financial frictions in the dynamics of macroeconomic and financial variables in Tunisia and examined the main transmission channels of shock propagation.

**Keywords:** bank leverage; financial frictions; DSGE model; financial stability; monetary policy

## 1. Introduction

The array of financial crises that have plagued economies around the world has demonstrated that economic performance is closely linked to disturbances in the financial sector. In developed countries, the banking system usually gathers financial markets to mobilize financial funds and stimulate economic growth. However, in developing countries, capital markets are not well developed, which puts banks' financial state at the heart of macroeconomic and financial dynamics. Hence, shocks originating in credit markets cause substantial output losses and large-scale unemployment (Gaies and Nabi 2021; Ngepah et al. 2022; Ali et al. 2023; Tabak et al. 2022; Zhang et al. 2023). Among the various origins of crises, including health, energy, and financial frictions, the latter remain particularly relevant. Financial frictions are the various restrictions that threaten the transmission of credit flows between banks and browsers, such as moral hazard and information asymmetry problems. Several studies on the role of financial frictions in shock propagation, such as those by Karmelavičius and Ramanauskas (2019), Zhang and Zhou (2021), Higgins (2023), Maxted (2023), Giakas (2023), and Chen et al. (2023), have been performed. Nevertheless, the literature concerning emerging and undeveloped markets is

limited; see Doojav and Kalirajan (2020), Francis et al. (2020), Akinci (2021), Ma and Jiang (2023), Chen et al. (2023), Giakas (2023), Gabriel et al. (2023).

Financial frictions can result in constraints on access to credit, higher financing costs, and information asymmetries. These frictions can affect banks' abilities to lend and invest in different types of assets. In the presence of financial frictions and economic shocks, banks can adjust their asset mix to mitigate risks or maximize their returns. For example, if financial frictions make lending more expensive or risky, banks may reduce their lending exposure and increase their reserves or investments in more liquid, less risky assets, such as sovereign securities. The composition of banks' liabilities affects their credit supply policies (Burietz et al. 2023; Moraux et al. 2023). Through the bank capital channel, an extension of the financial accelerator model devised by Bernanke et al. (1999), a decrease in financial asset prices reduces banks' net wealth via a loss in their portfolio of assets. In this context, the bank has two choices: either increasing its capital or reducing the credit supply. This last option seems easier to implement because increasing equity can be costly for weakly capitalized banks. Meh and Moran (2010) highlighted that banks' assets are only a combination of the loans of entrepreneurs, who can go bankrupt, while Gertler and Karadi (2011) highlighted that banks use their resources to buy risky assets. Hence, a decline in bank resources results, according to the Meh and Moran model, in the drying up of credit ("credit crunch"), and according to the Gertler and Kardi model, in the sale of risky assets. The presence of financial frictions amplifies the effects of these results.

Despite the several literature studies on the Tunisian banking system's vulnerability, research on the estimation of DSGE models for the Tunisian economy is scarce (Abdelli and Belhadj 2015; Chakroun 2019; Alimi and Chakroun 2021; Ben Salem et al. 2022). Alimi and Chakroun (2021) examine financial frictions and their effects on macroeconomic fluctuations in Tunisia using a Bayesian DSGE model. The model investigates the amplifying relationship between financial constraints and real economic shocks. Hamzaoui and Regaieg (2016). adopts the same methodology on the Tunisian economy but focuses on the effect of financial constraints on adjustments in the consumption and investment decisions of Tunisian economic agents.

Through our study, we offer two key contributions to the existing literature. First, this is, according to our best knowledge, the first study to highlight the effect of credit frictions on the conduct of monetary policy in the Tunisian economy. Second, our model highlights the crucial role of banking system vulnerability and bank leverage ratio in economic dynamics by introducing not only borrower–credit frictions but also lender–credit frictions.

Our empirical results offer valuable insights for policymakers and may aid them in making informed strategies and shaping market regulations. They provide valuable information about the channels through which volatility in the Tunisian banking sector with high bank leverage has important repercussions on real and financial volatility. The rest of the paper is organized as follows: Section 2 presents the description of DSGE models with financial friction. Section 3 presents the methodology adopted and the data used in the model simulation. Section 4 discusses the results found; finally, we will present the conclusion and the discussion.

## 2. Model Presentation

### 2.1. Households

We assume that the economy features a representative household that is infinitely lived and determine its consumption, C, and labor supply, l, to maximize its utility function as follows:

$$E_t \left\{ \sum_{i=0}^{\infty} \beta^i \left[ \ln(C_{t+i} - hC_{t+i-1}) - \frac{\nu}{1+\varphi} \left( l_{t+i}^s \right)^{1+\varphi} \right] \right\} \tag{1}$$

where $E_t$ is the expectation operator, $\beta^i$ is the household's subjective discount rate, h is the degree of internal habit formation, and $\varphi$ is the intertemporal elasticity associated with the labor supply.

The household saves by depositing funds in banks and buying government bonds at a nominal risk-free rate. This household is subject to the following budgetary constraints:

$$P_t C_t = w_t P_t l_{t+} P_t \text{prof}_t + (R_{t-1} B^T_{t-1} - B^T_t - P_t T)_t \tag{2}$$

The household receives a real wage, $w_t$, for supplying labor to retailers, d, and derives profit income from the ownership of retail firms and capital goods producers. $B^T_{t-1}$ are the financial assets (deposit) remunerated in period t with a nominal interest rate, $R_{t-1}$. These revenues are exploited in consumption $C_t$, payment taxes $T_t$, and investment in their financial assets.

### 2.2. Capital Goods Producers

Households produce their own capital goods by using technology and sell them to entrepreneurs at the currency price ($P_t Q_t$). The real expected profits of capital goods producers are then given by

$$E_t \left\{ \sum_{i=0}^{\infty} \frac{\varrho_{t+i}}{\varrho_t} \beta^i I_{t+i} \left[ Q_{t+i} \left( 1 - \frac{\eta_i}{2} \left( \frac{I_{t+i}}{I_{t+i-1}} - 1 \right)^2 \right) - 1 \right] \right\} \tag{3}$$

### 2.3. Retailers

The retailer is indexed by i and produces various products for consumption (Hafstead and Smith 2012). Each retailer operates under monopolistic competition and is owned by households; the product demand is given by

$$Y_t(i) = \left( \frac{P_t(i)}{P_t} \right)^{-\epsilon} Y_t \tag{4}$$

where $\epsilon > 1$ is the elasticity of substitution between different varieties. Retailers use the labor force ($lt(i)$) of households and rent capital services ($K^s_t(i)$) at a rental rate ($r^k_t$) from entrepreneurs. Hence, the production of retailer firm i is as follows:

$$Y_t(i) = (K^s_t(i))^{\alpha} (\exp(a_t) l_t(i))^{1-\alpha} \tag{5}$$

Retailers are subject to nominal rigidities in the form of Calvo (1983) contracts, which means that only a fraction of them, $1 - \xi^P$, are allowed to optimize their price in a given period.

### 2.4. Bankers

The model assumes that banks derive income from offering loans to nonfinancial firms. Bankers offer two types of credit: The first includes "risky inter-period loans", which are offered to entrepreneurs to purchase their capital stock in period t + 1. The second includes "risk-free inter-period working capital loans", Lrt (q), which are offered to retailers who used to pay for the labor and capital services dedicated to production at the end of period t.

Financial frictions are introduced in the model through the fact that after collecting deposits, the banker can divert a fraction of assets, $0 \leq \lambda \leq 1$, collected from the household, and declare bankruptcy (the moral hazard problem). In this case, the banker declares bankruptcy, and households recuperate residual assets.

$$V^b_t(q) = E_t \left\{ \sum_{i=0}^{\infty} (1-\theta)\theta^i \left( \frac{1}{\prod_{j=0}^{i} R^r_{t+1+j}} \right) N^b_{t+1+i}(q) \right\}, R^r_{t+1} = \frac{R_t}{\Pi_{t+1}} \tag{6}$$

The bank's net worth is

$$P_t N^b_t(q) = \left[ R^b_t P_{t-1} L^e_{t-1}(q) - R_{t-1} B_{t-1}(q) \right] \exp(e^z_t) \tag{7}$$

where $R_t^b$ is the net average return that the bank wins on the inter-period loan supply in period $t-1$ and $e_t^z$ is the exogenous capital shock. The model assumes that all banks choose an identical ratio between inter-period loans (loans to entrepreneurs) and their net worth. Therefore, $L_t^e = \varnothing_t^b N_t^b$, where $\varnothing_t^b$ is endogenous bank leverage. The variability $(\varnothing_t^b)$ is crucial for the results, and it represents the main parameter for total leverage, that is, the ratio of total loans to the bank's net worth $\left(L_t / N_t^b\right)$.

$$\widehat{\varnothing}_t^b = E_t\left\{\theta\beta^2 Z^2 \widehat{\varnothing}_{t+1}^b + \varnothing^b \frac{R^b}{R}\left(\widehat{R}_{t+1}^b - R_t\right)\right\} \tag{8}$$

The equation shows that bank leverage is positively related to the anticipated sum of profit margins on loan supply in period t and that $\widehat{R}_{t+1+i}^b - R_{t+i}$.

### 2.5. Entrepreneurs

At the end of period t, risk-neutral entrepreneur j buys capital $K_t^j$ for price $P_t Q_t$. The average return on capital across entrepreneurs is given by $R_t^K = \Pi_{t+1} \frac{r_{t+1}^k + Q_{t+1}(1-\delta)}{Q_t}$.

Similar to the bank's program, the model assumes that entrepreneurs die with a fixed probability of $1-\gamma$. Dying entrepreneurs consume their equity, Vt. This assumption ensures that entrepreneurs never become fully self-financed. The fraction $(1-\gamma)$ of entrepreneurs who have died is replaced by new entrepreneurs in each period, who receive a transfer. The households we considered were very small (Lubello et al. 2018), $N_t = \gamma V_t + W^e$.

### 2.6. Monetary Policy and Market Equilibrium

The central bank determines the risk-free interest rate by following the interest feedback rule in the following form:

$$R_t - 1 = (1 - \rho_i) + \psi_\pi(\log(\pi_t) - \log(\pi)) + \psi_y\left(\log(Y_t) - \log(\overline{Y_t})\right) + \rho_i(R_{t-1} - 1) + e_t^i \tag{9}$$

where $\rho_i$ represents the interest rate smoothing, and $\psi_\pi$ and $\psi_y$ are the coefficients associated with the deviation of inflation from its target and the output gap, respectively.

The equilibrium of our model is represented by the following equations:

$$
\begin{aligned}
S_t &= (1 - \xi^p)\left(\frac{\Pi_t}{\Pi_t^*}\right)^\varepsilon + \xi^p\left(\frac{\Pi_t}{\Pi_{t-1}^{\gamma_p}}\right)^\epsilon S_{t-1} \\
C_t^p &= C_t + C_t^e + C_t^b \\
Y_t &= S_t\left(I_t + C_t + \frac{R_t^k}{\Pi_t}Q_{t-1}K_{t-1}\mu\int_0^{\varpi_t}\omega f(\omega)d\omega\right) \\
Y_t &= (K_{t-1})^\alpha (A_t I_t)^{1-\alpha} \\
GDP_t &= I_t + C_t + G_t
\end{aligned} \tag{10}
$$

## 3. Data and Model Calibration

### 3.1. Data

In this study, we analyzed the effect of financial frictions on the propagation of real and financial shocks. We compared two alternative new Keynesian DSGE models, where the baseline is the model based on the passive banking sector devised by Bernanke et al. (1999), denoted here as "baseline model", and the other one is the model based on the bank leverage ratio devised by Rannenberg (2016), denoted here as "BFF model". Figure 1 describes the primary relationships among the economic agents.

We used the same quarterly dataset for both new Keynesian models, from 2000 Q1 to 2022 Q4. The quarterly data considered were real GDP, consumer price index, policy interest rate, credit, and credit spread. All series refer to the Central Bank of Tunisia (CBT) and the IMF database. The time trend in the data was eliminated for crucial variables

using the Hodrick–Prescott filter. Nonetheless, if boundaries were not accessible from the information, their worth was adjusted according to the comparative model instruments.

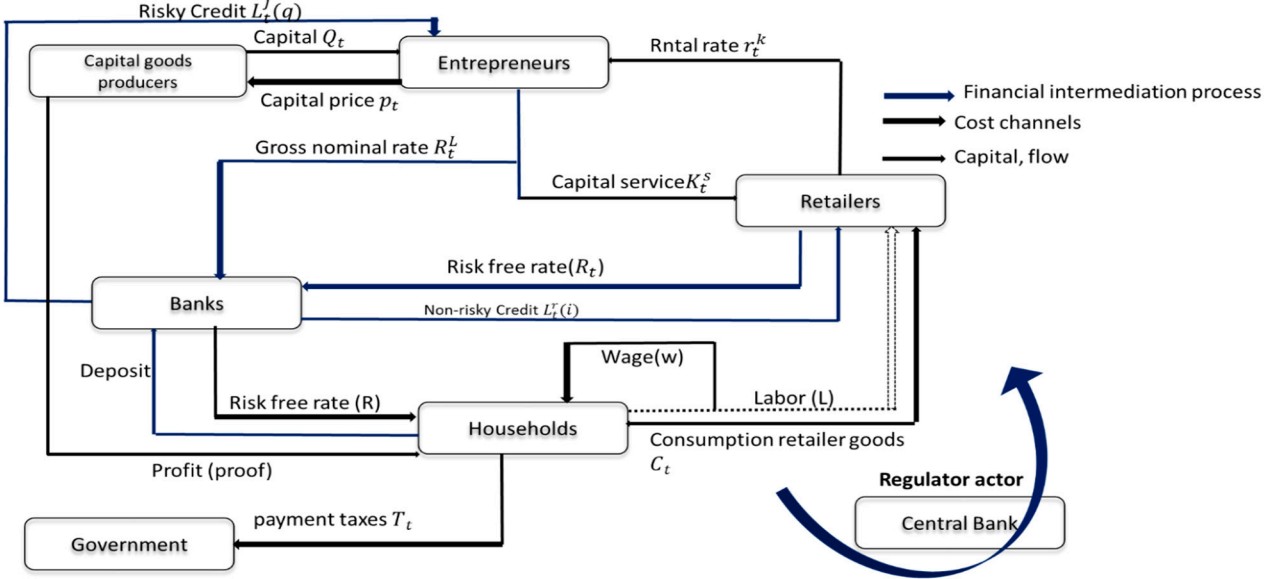

**Figure 1.** Overview representation of the DSGE model with financial frictions.

### 3.2. Calibration

Calibration is a crucial process to determine the values of unobservable parameters in the DSGE model. This method involves setting these parameters to specific values before estimating the DSGE model. To calculate the discount factor of households, β, we employed the same methodology as that used by Palić (2018), i.e., $r_{quarterly} = -\ln \beta$. The intertemporal discount rate was calculated such that the equilibrium interest rate on the Tunisian government bonds (treasury bond) was approximated by the long-term interest rate, denoted by R, which is a proxy for the risk-free rate. The quarterly rate was calculated by using the following equation: $r_{quarterly} = (1 + r_{annual})^{1/4} - 1$. In line with the previous equation, the corresponding quarterly rate was 1.665%. Hence, the discount factor was $\beta = e^{-r_{quarterly}} = e^{-\cdots} = 0.983$.

We now briefly detail the parameters related to the retailer's indices, $\xi_p$ and $\mu$. Price stickiness indices are among the principal differences between new Keynesian models and real business cycle models. According to Palić (2018), the average price stickiness is equal to $(1 - \theta)^{-1}$. Retailers should wholly pre-finance their capital and labor costs with working capital loans; hence, we set $\psi_k$ and $\psi_l = 1$. The capital elasticity of the output (α), the depreciation rate of capital (δ), and the elasticity of work disutility (φ) were fixed at 0.35, 0.025, and 0.25, respectively, as estimated by Alimi and Chakroun (2021).

The parameters relating to the various frictions in the banking and entrepreneurial sectors were calibrated such that the steady-state values of the key financial variables in the model matched their averages in the real data. Rannenberg (2016), Li and Wang (2020) and Mansour et al. (2021) also applied this methodology.

The bank leverage ratio was calibrated such that both debt and total assets matched the data. Based on quarterly data from 2000 to 2022 Q4, the average bank leverage ratio was 7.865. We calculated, for the same sample period, the bank capital ratio, $\frac{N^b}{L}$, with the average ratio between the bank's net worth available in the data stream database and the total debts. To match the data, this ratio was set to 5.694%.

### 3.3. Shocks and Prior Estimation

In total, there are 18 exogenous stochastic variables in the financial friction model. In the response function and moment comparison, we considered three stochastic processes: monetary policy shock, transitory productivity shock, and bank capital shock.

Exogenous shocks follow an AR (1) process of the type $\log(\varepsilon_{z,t}) = \rho_j \log(\varepsilon_{t-1}) + \epsilon_{z,t}$, where $\rho_j \in [0.1]$; $\epsilon_t$ is independent and identically distributed, with mean 0 and standard deviation equal to $\sigma^2$; and $\varepsilon = \{a, z, r\}$ identify productivity shock, bank capital shock, and monetary policy shock, respectively (Levieuge 2009).

Productivity shock $a_t$ was modeled as an autoregressive process of the ratio of gross value-added GVA in million dinars (TND) to the number of employed persons from 2000 Q1 to 2022 Q4. The AR coefficient of total factor productivity, $\rho_a$, was 0.9. The variance of the measurement errors, $\varepsilon_t^a$, was calibrated to correspond to 0.023. The volatility of total productivity was very low because new technologies are usually introduced slowly with respect to quarterly periods. Monetary policy shock was also modeled by using the autoregressive process of the money market interest rate. The persistent coefficient, $\rho_r$, was set to 0, with standard deviation ($\sigma_r$) equal to 0.022. the persistent coefficient of bank capital shocks, $\rho_z$, was 0.9, with standard deviation ($\sigma_z$) equal to 5%, according to Lubello and Rouabah (2018).

In Figure 2, we compare the prior and posterior distributions of the estimated standard deviation shocks and plot the average posterior distribution. As we can see, the data provided information pertinent to the estimation. For example, the standard deviation of the productivity shock had a prior of 0.08 and an estimated average of 0.09 from the posterior distribution. This result is close to the values found by Jouini and Rebei (2014) and Ben Salem et al. (2023). The bank capital shock showed a posterior distribution with an average of 0.0652, and it was fixed at 0.06.

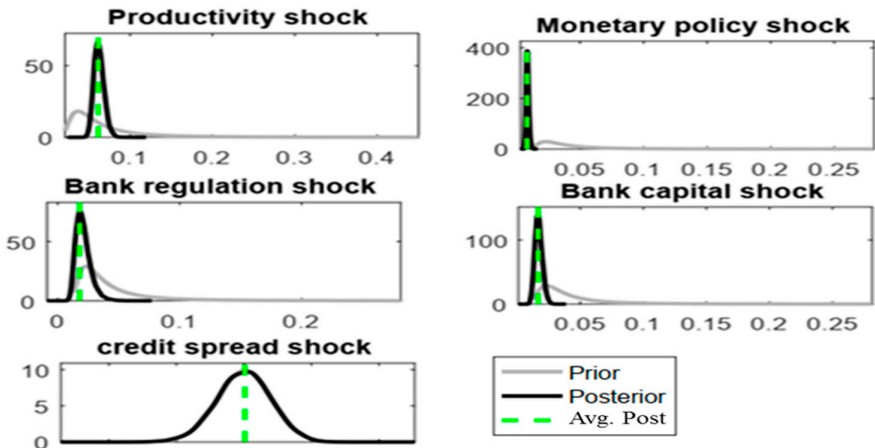

**Figure 2.** Prior and posterior distributions of structural shocks.

## 4. What Structural Shocks Drive the Tunisian Economy?

### 4.1. Restrictive Monetary Policy

The transmission of monetary policy shocks was analyzed by examining the Impulse Response Function, IRF of a 100 bps increase in the policy rate. We aimed to understand how financial frictions in the banking sector amplify the transmission of monetary policy. Every variable's reaction was expressed as the percentage deviation from its steady state, except for the rate variables, which were expressed as percentage points. Due to the various channels introduced in the model, we noted that the general effect of monetary policy on the transmission mechanism could be ambiguous; see Figure 3. This figure illustrates that the presence of financial intermediation led to amplified shocks. In the model with financial frictions (BFF), reductions in output, consumption, investment, entrepreneurs' net worth, and bank capital were more pronounced than in the passive bank model. Specifically,

monetary policy tightening in an economy with financial intermediation triggers a decline in capital prices, leading to an increase in the policy interest rate. Conversely, in Tunisia, where the loan rate is indexed to the policy rate, the loan costs increase, impacting firms' net value. This mechanism, commonly referred to as the Fisher effect, exerts a detrimental influence on investment and real economic activities (Le 2021). Furthermore, we found that the bank's net worth decreased in response to monetary tightening. Hence, our results are in accordance with the findings of Lamers et al. (2019), Harding and Klein (2022), and Choi and Choi (2021). This relationship can be explained by two reasons: On one hand, monetary tightening increases interest rates and makes credit more expensive for borrowers, thus reducing the demand for loans. Since Tunisian banks derive a large part of their income from the credit interest rate, their net capital decreases (Djebali and Zaghdoudi 2020). On the other hand, monetary tightening can also affect the profitability of banks by reducing the net interest margin (Kamara and Koirala 2023). Banks generally borrow in the short term and lend in the long term, and when short-term interest rates increase faster than long-term interest rates, it can reduce their profit margin (Mohamed 2020). However, we found that the response of loans to monetary tightening was mixed (Adrian and Liang 2018; Nguyen et al. 2022). Specifically, we found that loans for entrepreneurs first increased following the drop in their net worth, which forced them to increase the demand for external funds. However, bank capital erosion lowers a bank's ability to extend credit, which tightens its credit supply. This narrowing was found to be more critical in the BFF model, in which, after ten periods, it reached a lower level than that in the case of the frictionless banking sector (baseline model). The credit crunch decreased bank profitability after ten periods. If we consider the rational anticipation hypothesis, low profitability pushes households to withdraw their deposits, forcing banks to restrain their credit supply.

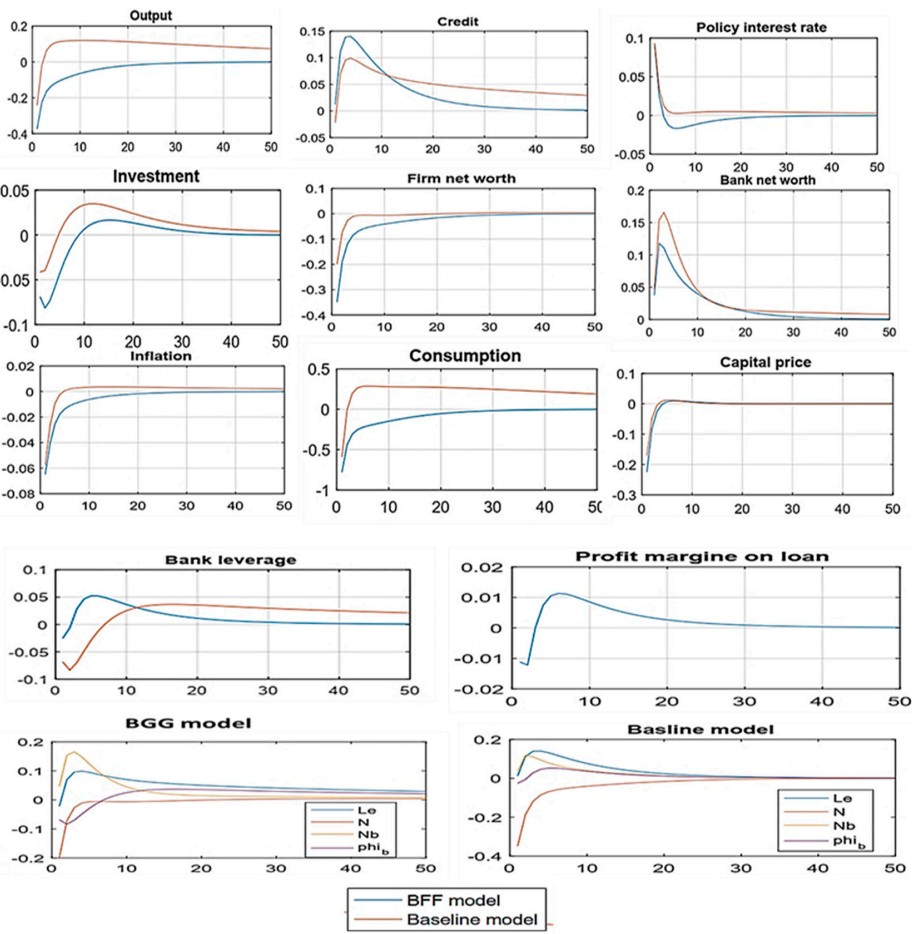

**Figure 3.** Monetary policy shock in DSGE models with and without financial frictions.

## 4.2. Adverse Productivity Shock

The transmission of productivity shocks was analyzed by examining IRFs by using the same set of models discussed in the previous subsection. Figure 4 shows a simultaneous decline in consumption and output during the initial ten periods in the baseline model, and these decreases were more pronounced than those seen in the model incorporating bank financial frictions. The occurrence of a negative productivity shock diminishes the financial standing of entrepreneurs, leading them to reduce investment. Consequently, this leads to a decreased level of capital and a reduction in entrepreneurs' net worth in the subsequent period, thereby exerting a negative impact on output, as outlined by Rubio (2020). In the BFF model, we found that the introduction of financial frictions in a DSGE model made the economy more sensitive to financial shocks, which resulted in increased inflation volatility compared with the model without these frictions; a negative productivity shock increased inflation by increasing firms' marginal costs and fostering an increase in credit supply along with risky credit—a scenario that was not observed in the baseline model (Gertler 2010). Consequently, inflation exhibits greater volatility in an imperfect banking sector than in an economy characterized by a passive banking sector. This is accompanied by increased fluctuations in output, investment, and consumption, particularly in the initial phases (Gallegati et al. 2019; Mohabatpoor et al. 2022).

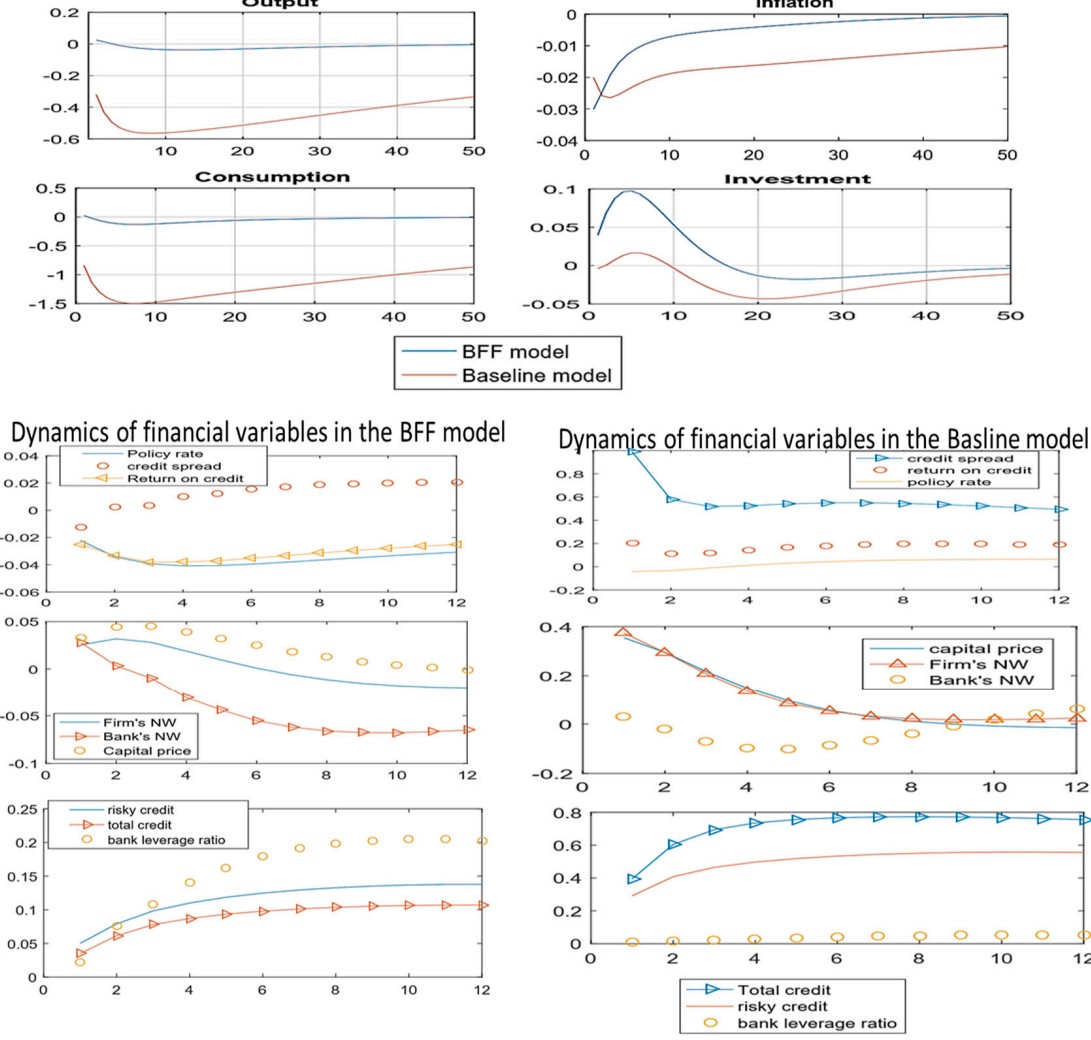

**Figure 4.** Adverse productivity shock in DSGE models with and without financial frictions.

Regarding the financial variables, in the BFF model, the impact of a productivity shock on economic variables was notably similar to that of a monetary policy shock. This similarity is explained by the interest rate instrument as follows: a negative productivity shock prompts the central bank to lower its policy interest rate, aiming to stimulate investments. However, due to the asymmetry of information in the Tunisian credit market, this reduction cannot be fully transmitted to the credit spread, posing a threat to the efficacy of monetary policy transmission. To maintain depositors' confidence, the bank increases its credit supply despite a decline in its capital, thus increasing the share of risky credit within its portfolio. This mechanism is effectively depicted in the figure. These simulation results indicate the presence of a moral hazard issue in Tunisian banks. Additionally, we found that during a period of productivity shock, the bank assumes higher risks in its credit portfolio, a phenomenon substantiated by the persistent increase in the bank leverage ratio.

### 4.3. Negative Bank Capital Shock and Conditions of Low and High Bank Leverage Ratios

We analyzed the persistent decrease in the bank capital ratio. To run the simulation, we introduced the possibility of an unexpected and persistent contraction in bank capital (Kb), as in Lubello and Rouabah (2018). This shock was calibrated such that it determined a decrease in bank capital of 5% on impact. In this exercise, we used the bank leverage ratio as proxy of the measure of financial frictions, whereby the more the ratio increased, the more banks took risks (Suh and Walker 2016; Li 2022; Le 2021). Three cases were considered: a high level of financial frictions, $\widehat{\varnothing}_t^b = 15\%$; an average level of financial frictions, $\widehat{\varnothing}_t^b = 8.9\%$; and a low level of financial frictions, $\widehat{\varnothing}_t^b = 3\%$. Figure 5 presents the IRF of the variables for a negative bank capital shock and for different values of financial friction. We found that a lower leverage ratio led to a significant increase in the lenders' interest rate and a slower decrease in net worth. This can be explained by the fact that a lower leverage ratio indicates a more prudent use of debt by the bank. Hence, to offset the reduced debt risk, the bank could increase the credit interest rates. The slower decline in net worth is due to the bank taking fewer risks, which can lead to a more moderate but stabler growth in net worth. Furthermore, we noted that a low leverage ratio increased the spread between global loan rates and global deposit rates. This resulted in a higher profit margin on loans, although this increase was not as proportionate as in the case of the high leverage ratio. This dynamic is explained by the fact that a low leverage ratio indicates a more prudent use of debt, which can lead to relatively lower borrowing costs. As a result, the bank can maintain higher interest rates on loans relative to the rates it pays on deposits, thereby increasing its profit margin Giakas (2023). When banks are well capitalized, they can overcome financial frictions and are capable of channeling funds from less productive agents to more productive ones, which positively influences the GDP (Burietz et al. 2023). When banks are less capitalized, there are reductions in GDP, investment, consumption, and bank capital. In this case, the capital price progressively adapts to the credit crunch supply. In this model, a bank capital shock might be similar to a demand shock, in that it decreases both output and inflation (Rannenberg 2016). Negative bank capital with low financial frictions pushes banks to utilize bankers' limited wealth. This low leverage has an expected impact on the cost of equity funding (owing to its shortage in the short term) and on the pattern of the collective wealth of bankers (in the medium to long term). Applying an excessive effort to increase capital (to assure bank capital requirements) reduces the supply of loans due to the partial ability of Tunisian banks to meet the regulation constraint only by increasing the filtering effort, Labus and Labus (2019).

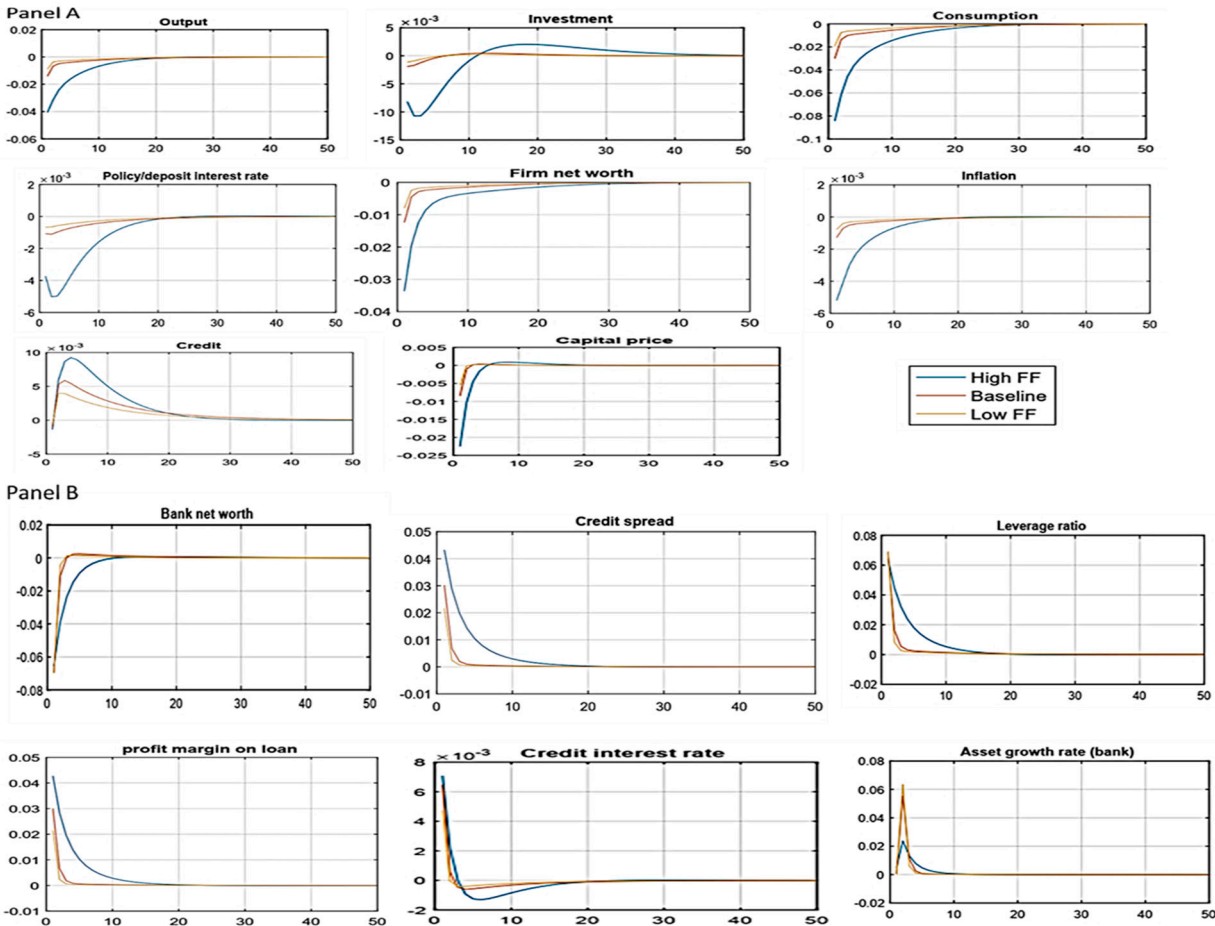

**Figure 5.** IRF of negative bank capital shock in the context of different financial frictions.

## 5. Model Evaluation

Model evaluation includes selecting a loss function that evaluates the distance between the real indicators and the result of the simulated indicators. Canova (2005) identified four groups of model validation approaches: first, those based on the R2-type measure, which is included to provide an evaluation of the estimation; second, those based on the sampling volatility of the real data; third, those based on the sensibility test of the calibrated parameters according to their volatility, which allows for the fitting of the range between moments and the results from the DSGE model (to assess DSGE models, Soderlind (1994) used this approach); fourth, those that evaluate the distance by utilizing the sampling volatility of both real and simulated variables. The first three models assume stochastic shocks and parameters to be calibrated according to the literature. It is conceivable to differentiate approaches that take into consideration volatility in the parameters but not in the exogenous shocks. An alternative methodology investigated by several other researchers, DSGE-VAR, is based on the use of the VAR result as an econometric instrument for empirical validation, combining prior information of the DSGE model in the estimation.

In this study, in accordance with Afrin (2017), we compared the robustness of the theoretical moment generated by the DSGE model with that generated by the real data. To measure the relative importance of each process shock, we adopted a historical decomposition methodology, in line with Wong (2017).

## 5.1. Relative Fit of DSGE Model with Alternative Bayesian Facto

To compare the two DSGE models, we differentiated between the two marginal likelihood functions as follows:

$$Log\left(BF_{ij}\right) = \log(p(Y \backslash M_i)) - \log(p(Y \backslash M_j))$$

where $p(Y \backslash M_i)$ and $p(Y \backslash M_j)$ are the marginal data densities of the models with and without financial frictions, respectively.

Another method, proposed by Kass and Raftery (1995) and validated by Merola (2014), involves the use of the KR criterion. This method supposes that if the value of 2 log above 10 makes model "i" very effective, if the value is between 6 and 10, there is strong evidence for model "i". If the value is between 2 and 6, there is positive evidence for model i, while if the value is between 0 and 2, model "i" is ineffective. It should be noted that for the sake of comparison, it is necessary that the two models be estimated using the same sample, variables, and shocks (Merola 2014). Table 1 shows the result. We found that the model performance improved when the friction mechanism between entrepreneurs and the bank was introduced. The marginal likelihood or the marginal data density improved by 6 log points. Therefore, according to the KR criterion, there is strong evidence in favor of the model with financial frictions. This means that the model introducing frictions between lenders and borrowers represents a better model for Tunisian policymakers than the model only introducing frictions one-sidedly.

**Table 1.** Model comparison based on marginal data density.

| Estimation Sample | log($p(Y \backslash M_i)$) | log($p(Y \backslash M_j)$) | Kass and Raftery Criterion (2log (BF)) |
|---|---|---|---|
| 2000 Q1 to 2022 Q3 | −314 | −320 | 12 |

## 5.2. Absolute Evaluation: Theoretical Moment Comparison

Table 2 presents the posterior predictive examination, in which the moments in the model produced artificial data rather than inaccurate data.

**Table 2.** Theoretical moment comparison: posterior estimation analysis.

| Variable | S.D Baseline Model | S.D BFF Model | SD Real Data | AC (1) Baseline Model | Ac(1) BFF Model | Ac (1) Real Data | AC (2) Baseline Model | AC (2) BFF Model | AC (2) Real Data |
|---|---|---|---|---|---|---|---|---|---|
| **Output** | 7.325 | 1.66 | 5.926 | 0.9912 | 0.844 | 0.770 | 0.9793 | 0.7625 | 0.539 |
| **Consumption** | 18.188 | 2.60 | 0.523 | 0.9919 | 0.851 | 0.968 | 0.9817 | 0.7789 | 0.935 |
| Investment | 1.3256 | 1.14 | 0.626 | 0.9803 | 0.976 | 0.925 | 0.9352 | 0.9247 | 0.848 |
| **Inflation** | 0.2195 | 0.688 | 0.9779 | 0.9317 | 0.824 | 0.725 | 0.8881 | 0.7136 | 0.693 |
| Interest rate | 0.3076 | 0.584 | 0.6751 | 0.9228 | 0.766 | 0.767 | 0.8813 | 0.6517 | 0.746 |
| Firm's loan | 2.7938 | 1.615 | 1.4525 | 0.9956 | 0.984 | | 0.9879 | 0.9588 | 0.912 |
| Loan | 3.7697 | 1.29 | 0.9343 | 0.9930 | 0.944 | 0.986 | 0.9833 | 0.8776 | 0.971 |
| Leverage ratio | 1.3341 | 3.33 | 2.13 | 0.9889 | 0.969 | 0.989 | 0.9703 | 0.9511 | 0.978 |
| Bank capital | 0.8257 | 1.0024 | 0.9223 | 0.9875 | 0.964 | 0.984 | 0.9325 | 0.9471 | 0.968 |
| Credit spread | - | 0.737 | 1.468 | 0.9099 | 0.524 | 0.305 | 0.8976 | 0.5284 | 0.236 |
| Moral hazard | 0.000 | 0.764 | 0.683 | 0.000 | −0.076 | −0.042 | 0.000 | −0.088 | −0.183 |

The standard deviations (SDs) of the models demonstrated mixed performance. The model with financial frictions was quite acceptable in reproducing the volatility of policy interest rate, inflation, credit to entrepreneurs, total credit, bank leverage ratio, credit spread, and moral hazard problem, but it moderately overpredicted investment output and consumption. Therefore, compared with the baseline model, the BFF model performed satisfactorily in reproducing the volatility of most variables, which matched the standard volatility in the data. Autocorrelation coefficients up to the second order were calculated for the two models and are here shown in the right panel of the table. As we can see, the models replicated the

persistence in the data very closely for both financial and actual variables. Therefore, both the BFF model and the baseline model can be said to be autocorrelated (Lyu et al. 2023).

### 5.3. Variance Decomposition

The contributions of every structural shock to the forecast error variance of the macroeconomic and financial variables over the horizons of the 12 periods for the two models are shown in Table 3.

**Table 3.** Forecast error variance decomposition (in %) for both models.

| Shock | GDP | Consumption | Investment | Inflation | Policy Rate | Risky Credit | Total Credit | Bank Leverage | Credit Spread |
|---|---|---|---|---|---|---|---|---|---|
| **DSGE model with financial friction s** | | | | | | | | | |
| Monetary policy shock | 51.42 | 46.90 | 8084 | 42.31 | 36.68 | 25.71 | 2.86 | 22.66 | 1.93 |
| Productivity shock | 27.57 | 43.02 | 53.21 | 43.67 | 48.10 | 49.90 | 22.63 | 46.56 | 10.85 |
| Bank capital shock | 18.62 | 9.19 | 37.64 | 13.37 | 14.52 | 23.45 | 70.86 | 29.86 | 27.84 |
| **DSGE model without financial friction s** | | | | | | | | | |
| Monetary policy shock | 1.34 | 1.35 | 1.21 | 9.63 | 11.72 | | 1.90 | | 3.75 |
| Productivity shock | 95.6 | 95.52 | 96.42 | 82.02 | 85.41 | | 94.98 | | |
| Bank capital shock | 18.62 | 9.19 | 37.64 | 13.37 | 14.52 | 23.45 | 70.86 | 29.86 | 27.84 |

The table shows that financial frictions and bank capital shocks played a more substantial role in the dynamics of the financial variables than the other types of shock. At the same time, their impact on the baseline model was weak. The variance decomposition showed that the monetary policy shock explained 51.4% of the variations in the BFF model and approximately 1.3% of those in the baseline model. This is because the monetary strategies adopted by the Central Bank of Tunisia play an essential role in GDP dynamics when the banking sector is imperfect. Simultaneously, the impact deteriorates in the context of a passive credit market.

The bank capital shock explains 18.6% of the GDP variance, 9.2% of the consumption variance, 38% of the investment variance, and 13.3% of the inflation variance. Hence, the financial situation of the Tunisian banking sector influences the average macroeconomic dynamics, which provides evidence of the endogenous bank capital channel. Furthermore, the bank capital shock affects 28% of the credit spread volatility and 30% of the total credit variation. Most importantly, we found that 71% of the leverage ratio variation is explained by bank capital, which proves the effect of bank capital on risk taken by banks. This result confirms the conclusions obtained by Rannenberg (2016) and Higgins (2023), who showed that bank capital and bank leverage have an important effect on economic dynamics.

In the BFF model, we found that inflation volatility was mainly determined by the policy rate, a record of approximately 43%. This decomposition analysis shows that monetary policy orientation stabilizes inflation, which has been the main objective of the Central Bank of Tunisia since 2006. In contrast, this correlation was not verified in the baseline model. Hence, the model with frictions in the financial sector appears to match the reality of the Tunisian economy.

### 6. Discussion and Conclusions

In conclusion, the findings of our study reveal significant insights into the impact of financial intermediation and frictions on the economy, particularly in the Tunisian context. The presence of financial intermediation appears to amplify shocks, as evidenced by the more pronounced reduction in output, consumption, investment, entrepreneurs' net worth, and bank capital found in the model with financial frictions (BFF) compared with the passive bank model. One noteworthy observation is the decrease in the bank's net worth in response to monetary tightening, which is in line with previous research (Lamers et al. 2019; Choi and Choi 2021; Harding and Klein 2022). This can be attributed to the dual

effects of increased interest rates, i.e., credit becoming more expensive for borrowers and the potential reduction in the net interest margin, affecting banks' profitability. However, the response of loans to monetary tightening appears to be mixed, with initial increases for entrepreneurs followed by a subsequent tightening of credit supply due to bank capital erosion, especially in the BFF model.

By analyzing the impact of productivity shocks, we found that the introduction of financial frictions made the economy more sensitive to such shocks. This heightened sensitivity led to increased inflation volatility, stemming from firms' increased marginal costs and an increase in credit supply, which was not observed in the baseline model. The imperfect banking sector resulted in greater fluctuations in inflation, output, investment, and consumption, particularly in the initial phases. This is explained by the direct effect of financial frictions on marginal costs for companies. When productivity declines due to a negative shock, firms may find it difficult to absorb these additional costs, resulting in higher selling prices. By examining negative bank capital shocks, we found that a lower leverage ratio contributed to a significant increase in lender interest rates and a slower decrease in net worth. Well-capitalized banks proved to be more resilient, overcoming financial frictions and positively influencing the GDP by channeling funds from less productive agents to more productive ones. In contrast, having less-capitalized banks led to reductions in GDP, investment, consumption, and bank capital, resembling a demand shock, which impacted both output and inflation.

The model evaluation exercise demonstrated a notable improvement in the marginal likelihood or marginal data density when financial frictions for both entrepreneurs and banks were introduced. According to the Kass and Raftery criterion, this improvement provides strong evidence in favor of the model with financial frictions, suggesting that it represents a better framework for Tunisian policymakers than the model without financial frictions. This implies that considering financial frictions in economic modeling can offer more accurate insights and guidance for policy decisions in the Tunisian context. These findings are important, especially since the Central Bank of Tunisia currently does not have a specific DSGE model. Therefore, the CBT should take heed of the frictions within the banking sector when establishing its DSGE model in the future. The comparative analysis of both DSGE models showed that performance improved when the model included the financial friction mechanism.

Based on the analysis of the forecast error variance decomposition, it was evident that within the context of explicit financial frictions, shocks to bank capital played a very significant role in shaping the dynamics of financial variables. Conversely, their impact was less pronounced in the model featuring a passive banking sector. Additionally, our findings indicate that capital variation was responsible for 71% of the changes in the leverage ratio. This implies that as banks become more capitalized, they tend to take on more risks in their credit supply activities.

Our results offer important implications for economic decision-makers: First, models with financial frictions make it possible to identify and analyze the channels through which monetary and financial policies affect the economy. This can help policymakers design more targeted and effective policies to achieve macroeconomic objectives, such as stabilizing inflation, promoting economic growth and financial stability. Second, models with financial frictions more faithfully capture the imperfections and specific dynamics of the Tunisian financial system. Therefore, they can offer a more accurate representation of how monetary and macroprudential policies affect the economy as a whole. By using more sophisticated models with financial frictions, policymakers can develop long-term strategies to promote sustainable and inclusive economic growth. This involves taking into account the complex interactions between monetary policies, financial regulations and other economic and social factors.

**Author Contributions:** S.B.S.: Conceptualization, methodology and writing—original draft preparation. M.L.: Validation, formal analysis and data curation. S.S.: Visualization and funding acquisition. All authors have read and agreed to the published version of the manuscript.

**Funding:** This research was funded by Saudi Electronic University, 8683 wadi al hilali 4256, Al aziziyah Dist, Jeddah 23338, Saudi Arabia.

**Institutional Review Board Statement:** Not applicable.

**Informed Consent Statement:** Not applicable.

**Data Availability Statement:** All data are taken from the database of the National Institute of Statistics (INS), www.ins.tn, and the site of the Central Bank of Tunisia (BCT), https://www.bct.gov.tn/bct/siteprod/tableau_statistique.jsp?params=PL203030&prov=1 (accessed on 29 February 2024).

**Conflicts of Interest:** The authors declare no conflicts of interest.

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
