# Peer review of "Effect of Financial Frictions on Monetary Policy Conduct: A Comparative Analysis of DSGE Models with and without Financial Frictions"

_economies, doi:10.3390/economies12030072_

Round 1

Reviewer 1 Report

Comments and Suggestions for Authors

Please see the attached form.

Comments on the Quality of English Language

The exposition needs to be significantly improved.

Reviewer 2 Report

Comments and Suggestions for Authors

I think the work may be worthy of publication, but several changes need to be made. None of them are very important, but I consider them necessary. I list them in order of appearance:

1.       In the abstract, the reader suddenly reads the Tunisian case. In other words, he or she must assume that an empirical analysis will be carried out for the Tunisian case.

2.       The way in which quotations are included must be improved. It is never done this way, and it is something that "stains" the reading.

3.       Please elaborate a bit on the link between the first and the second paragraphs of the Introduction.

4.       “BGG’s (1999)”. If you are not an expert, you do not understand. Please include the citation.

5.       I miss the structure of the paper at the end of the Introduction. It always helps the reader.

6.       What does “v” denote in equation (1.1)?

7.       in the second equation the symbol + is not a subscript

8.       I would enumerate all equations and with a single digit. Why 1.1?

9.     Third equation. What is ?

10.   Fifth equation. What does “a” denote? It can be assumed, but you have to indicate it.

11.   In sum, please pay attention to the presentation of the model and equations. I do not continue.

12.   Page 5. First paragraph. I would like to know something on the sensitivity of the results?

13.   Second paragraph. I agree, but inform the reader. “some parameters” is not a good indication.

14.   AR(1). I know this is the usual assumption about the behaviour of shocks, and I have used it myself. But why not for example an AR(2). Could you say more about this?

15.   Did you think on the number of hours worked to compute productivity? The period is so long that, although I do not know the case study, I imagine that changes in the labour market have been important and, if so, considering hours worked rather than number of workers, provided there is data, might have been advisable. There is quite a lot of evidence on this.

16.   Do all readers know the meaning of IRF? It is convenient to write it in layman's terms.

17.   BGG in Figure 3?

18.   Baseline model and not Basline in Figure 4.

19.   For consistency in this figure, why not keep the order, and the type of line to the left and right of the figure? Why not place the legend of the last figure like the rest?

20.   Figure 5. Maybe is better average than baseline?

21.   From section 5 onwards the wording of the article worsens. Not only that, there are several typos that I will not list. Attention should be paid to this comment.

22.   A clear example is the first paragraph, which is rather confusing or, at least, could be improved.

23.   Page 11. Line 351. Model “i”.

24.   Page 12. Line 379. Table 3, not Table 4.

25.   Remember what you said about policymakers at the end of the Introduction. This is precisely the best way to close the paper. Please make an effort.

26.   Be consistent with the style used in the References section.

Reviewer 3 Report

Comments and Suggestions for Authors

This paper examines the effects of bank leverage and financial friction on the transmission of real and financial shocks in the Tunisian economy. Therefore, this paper calibrates a DSGE model (BFF model) that assumes financial friction and a DSGE model (Base line model) that does not assume financial friction, and compares their performance. As a result, the authors found that the BFF model was more successful in reproducing the dynamics of the Tunisian economy. Financial friction is an important issue not only in developed countries but also in developing countries, and the contribution of this paper, which quantitatively explores its effects, is highly appreciated.

However, there are major problems with the format of the paper, and as it stands, it is not up to the standard for publication. First, in the explanation of the model in Section 2, only one equation is numbered, and explanations for many variables are missing. Second, many of the papers cited in the paper are not listed in the references, and the format of citations in the references is inconsistent (furthermore, [25], [26], [27] in the references all refer to the same paper).

On the other hand, although the model in this paper seems to assume a closed economy, shocks from abroad are also thought to play an important role in an economy like Tunisia. It would be desirable for the authors to express their views on this point.

Round 2

Reviewer 1 Report

Comments and Suggestions for Authors

The exposition is greatly improved.  The paper is a valuable contribution.